# Red light-transmittance bagging promotes carotenoid accumulation of grapefruit during ripening

Xiulian Huang[1,3], Linping Hu[1,3], Wenbin Kong[2], Can Yang[1] & Wanpeng Xi [1✉]

Light, a crucial environmental signal, is involved in the regulation of secondary metabolites. To understand the mechanism by which light influences carotenoid metabolism, grapefruits were bagged with four types of light-transmitting bags that altered the transmission of solar light. We show that light-transmitting bagging induced changes in carotenoid metabolism during fruit ripening. Compared with natural light, red light (RL)-transmittance treatment significantly increases the total carotenoid content by 62%. Based on weighted gene co-expression network analysis (WGCNA), 'blue' and 'turquoise' modules are remarkably associated with carotenoid metabolism under different light treatment ($p < 0.05$). Transcriptome analysis identifies transcription factors (TFs) bHLH128, NAC2-like/21/72, MYB-like, AGL11/AGL61, ERF023/062, WRKY20, SBPlike-7/13 as being involved in the regulation of carotenoid metabolism in response to RL. Under RL treatment, these TFs regulate the accumulation of carotenoids by directly modulating the expression of carotenogenic genes, including *GGPPS2*, *PDS*, *Z-ISO*, *ZDS2/7*, *CRTISO3*, *CYP97A*, *CHYB*, *ZEP2*, *CCD1-2*. Based on these results, a network of the regulation of carotenoid metabolism by light in citrus fruits is preliminarily proposed. These results show that RL treatments have great potential to improve coloration and nutritional quality of citrus fruits.

[1] College of Horticulture and Landscape Architecture, Southwest University, Chongqing 400716, China. [2] Chongqing Agricultural Technology Extension Station, Chongqing 401121, China. [3] These authors contributed equally: Xiulian Huang, Linping Hu. ✉email: xwp1999@zju.edu.cn

Carotenoids are a large class of natural lipid-soluble pigments that are extensively distributed in plants and play important roles in plant growth and development, being involved in photosynthesis, photomorphogenesis, photoprotection, and phytohormone synthesis[1,2]. The accumulation of carotenoids confers on many fruits and vegetables their various colors, such as yellow, orange, and red hues[3]. In addition, their catabolites provide precursors for the synthesis of abscisic acid (ABA) and strigolactones (SLs), which participate in various biological processes and stress responses[4]. In humans, carotenoids in plant-based foods are an important source of dietary vitamin A, which is essential for health and nutrition, and carotenoid-rich diets are correlated with a significant reduction in the risks of chronic diseases such as cancers, cardiovascular diseases, and several degenerative diseases[2,5]. The concentrations of carotenoids in fleshy fruits thus greatly influence their commercial and nutritional value.

The grapefruit (*Citrus paradisi* Macf.) is an economically important tropical cultivated citrus fruit[6]. In 2018, the planted area of grapefruit in China was about 9200 hectares, about 25% of the global planted area, while annual production (around 5 million tons) accounted for approximately 54% of global output, indicating that grapefruit is an important part of China's citrus production (FAO statistics, http://www.fao.org/home/en/). Red grapefruit is becoming more and more preferred by consumers for its unique flavor and attractive pulp color. Besides having an abundance of a wide variety of health-promoting compounds such as flavonoids, dietary fiber, and vitamin C[7], grapefruits are richer in carotenoids than other citrus species and thus represent an ideal material for investigating carotenoid metabolism.

The pathway of carotenoid biosynthesis has been clearly established in plants[8]. The five-carbon prenyl diphosphate isopentenyl diphosphate (*IPP*) and its double-bond isomer dimethylallyl diphosphate (*DMAPP*) are synthesized in plastids via the 2-C-methyl-D-erythritol 4-phosphate (MEP) pathway. The subsequent condensation of two molecules of geranylgeranyl diphosphate (*GGPP*), produced from *IPP* and *DMAPP*, by phytoene synthase (*PSY*) generates the colorless 15-*cis*-phytoene. After sequential desaturation and isomerization reactions catalyzed by phytoene desaturase (*PDS*) and ζ-carotene desaturase (*ZDS*), ζ-carotene isomerase (*Z-ISO*) and carotenoid isomerase (*CRTISO*), respectively, phytoene is converted into the red all-*trans*-lycopene[9]. The production of α- and β-carotene from lycopene involves a set of cyclization reactions catalyzed by lycopene ε-cyclase (*LCYE*) and lycopene β-cyclase (*LCYB*) or *LCYB* alone, representing the β, ε- and β, β-branches of the pathway, respectively, and this step is the pivotal branch point in carotenoid metabolism. Next, α-carotene is converted into lutein by β-ring hydroxylase (*CYP97A*) and ε-ring hydroxylase (*CYP97C*) of the cytochrome P450 family. The production of zeaxanthin from β-carotene is catalyzed by β-carotene hydroxylase (*CHYB*), and violaxanthin is generated via antheraxanthin by zeaxanthin epoxidase (*ZEP*). The cleavage of carotenoids is catalyzed by the proteins of carotenoid-cleavage genes (*CCD* or *NCED*), producing apocarotenoids such as β-ionone, β-citraurin, and ABA[1,3].

Carotenoid biosynthesis and degradation are coordinated by a range of enzymes encoded by structure genes and transcription factors (TFs)[8]. These structure genes have been identified and isolated in many plant species to date[3,8]. However, only a few transcription factors related to carotenoid metabolism have been identified in plants, including RIPENING INHIBITOR (RIN) and FRUITFULL1/2 (FUL1/2) in the MADS-box family; PIF1, TOMATO AGAMOUS LIKE1 (TAGL1), SlMADS1, SlNAC1/4, FcrNAC22, SlAP2a, SlERF6, and SlBBX20 in tomatoes[10–13]; CsMADS5/6 and CrMYB68 in oranges (flavedo)[14–16]; CpEIN3a, CpNAC1/2, CpbHLH1/2 and Cp SBP1 in papaya[17–20]; AdMYB7

in kiwifruits[21]; and R2R3-MYB subgroup Reduced Carotenoid Pigmentation 1/2 (RCP1/2) in monkeyflower species[12,22]. Compared with that of anthocyanin metabolism, the transcriptional regulation of carotenoid metabolism is far from understood.

Light not only provides the energy required for photosynthesis but also as a crucial environmental signal to participate in the regulation of a variety of metabolic processes during plant development[23,24]. An increasing number of investigations suggest that light signals also play a fundamental role in secondary metabolism in fruit. However, the majority of studies have focused on the effect of postharvest light treatment on fruit quality, with only several works referring to the impact of developmental light treatment on fruit quality[25]. As an effective method of protecting fruit from insect infestations, bird attack, and sunburn as well as reducing disease incidence rate and chemical residues, fruit bagging is extensively used in modern orchards[26]. Light-transmitting paper bags of different colors can absorb the light waves of the corresponding colors, making their use a feasible approach for investigation of how light influences phytochemicals metabolism during fruit development.

This study was carried out to know the role of light quality on the carotenoid accumulation in grapefruit and understand the transcriptional regulatory mechanism underlying light signals during fruit ripening. The carotenoid level of 'Huoyan' grapefruit pulp treated with different light-transmittance during the ripening were compared, WGCNA were employed to identify the key genes and TFs responsible for carotenoid metabolism during the process. Based on these results, a regulatory network of carotenoid metabolism in response to red-light was preliminarily proposed. These findings provide new insight into carotenoid metabolism and demonstrate a potential approach for improvement of the coloration and nutritional quality of citrus fruit and other horticultural crops.

## Results

**Effects of light transmittance on TSS, TA, and CCI during fruit ripening.** Compared with that in the control group (CK), the TSS content of the grapefruits treated with RL, BL and WL gradually increased during fruit ripening and were significantly higher than that in CK treatment at 220 DAB ($p < 0.05$) (Fig. 1a, Supplementary Data 1). The effect of light treatment on TA content is similar, and TA content in RL, BL and WL treatments were all reduced (Fig. 1b). It is worth noting that the TA content under BL treatment was significantly decreased compared to the CK group. CCI were gradually increased in all light treatments and CCI for all light treatments were remarkably higher that in the control group (Fig. 1c).

**Effects of light transmittance on carotenoid accumulation during fruit ripening.** Five major carotenoids were identified from 'Huoyan' grapefruit pulp, including β-carotene, phytofluene, ζ-carotene, lycopene, and 9-*cis*-violaxanthin (Supplementary Table 1). The carotenoid profiles differed in the four light treatments, and the 'Huoyan' grapefruit pulp was rich in β-carotene and lycopene (Fig. 2, Supplementary Data 2). Compared with the control group (CK), the total carotenoid content was the highest in the grapefruits treated with RL (1.62-fold), followed by the grapefruits treated with WL (1.08-fold) and BL (1.01-fold) at 220 DAB. Subsequently, we found that the content of β-carotene, ζ-carotene, lycopene and phytofluene were gradually increased as fruit ripening under different light treatments while 9-*cis*-violaxanthin content didn't show marked changes. Compared with CK, the β-carotene, lycopene and ζ-carotene content were significantly increased under RL treatment and remarkably higher than those in BL and WL treatment at 220 DAB.

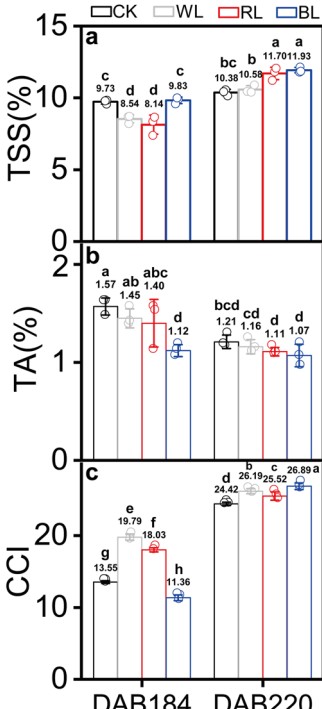

**Fig. 1 Effect of grapefruit bagged with four light-transmitting bags on the TSS content, TA content and CCI of grapefruits during ripening. a–c** show effect of grapefruit bagged with four light-transmitting bags on the TSS content (**a**), TA content (**b**) and CCI (**c**) of grapefruits during ripening. Error bar indicate standard error from three biological replicates ($n = 3$). CK: control group; WL: white-light treatment; RL: red-light treatment; BL: blue-light treatment. DAB184 and 220 represent 184 and 220 days after blossom, respectively. Different letters indicate statistically significant difference in one-way ANOVA analysis ($p < 0.05$). The circle represents the individual data point derived from three independent experiments.

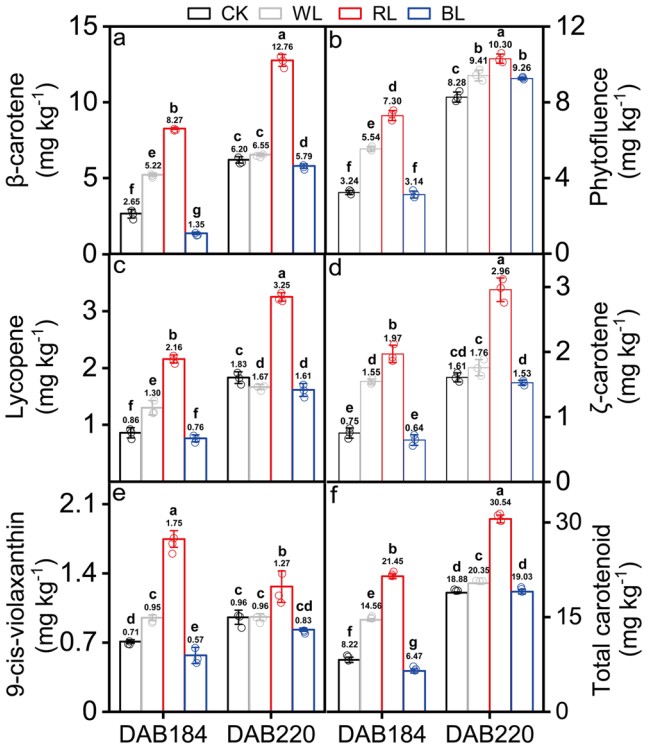

**Fig. 2 Effects of different light treatments on carotenoid content during grapefruit ripening.** Effects of different light treatments on β-carotene (**a**), phytofluene (**b**), lycopene (**c**), ζ-carotene (**d**), 9-*cis*-violaxanthin (**e**), and total carotenoid content (**f**) at DAB 184 and 220. Error bars indicate the standard error from three biological replicates ($n = 3$). CK: control group; WL: white-light treatment; RL: red-light treatment; BL: blue-light treatment. DAB184 and 220 represent 184 and 220 days after blossom, respectively. Different letters indicate statistically significant difference in one-way ANOVA analysis ($p < 0.05$). The circle represents the individual data point derived from three independent experiments.

**Transcriptome profiles during fruit ripening.** RNA integrity number (RIN) for all RNA samples and RIN score were evaluated and ranged from 7.8~9.2 (Supplementary Table 2) and all of samples meets the requirements of library construction and sequencing. The average clean reads number of mRNA libraries for twenty-four samples ranged from 42.58 to 45.46 million (Supplementary Table 3). The alignment of the clean reads against the reference genome and reference gene sequences generated a total of 22176 unigenes (Supplementary Table 4). In four light treatments, the median of gene expression level ranged from 1.05 to 1.15 and there were significant differences between the samples (Supplementary Figure 1). The fruit samples treated with BL showed the lowest (1.05) and biggest median (1.15), respectively, at 184 and 220 DAB. However, in the RL treatment groups, the gene expression levels for samples were relatively stable (from 1.10 to 1.11) during ripening.

**Identification of differentially expressed genes (DEGs).** Based on RNA sequencing results, a total of 6088 DEGs were identified during fruit ripening, with 4514, 2450 and 3096 DEGs showing differential expression between CK and RL, DS and BL, and DS and WL, respectively (Fig. 3a, Supplementary Data 3). In the RL group, 3248 and 1810 DEGs were identified at 184 and 220 DAB, respectively, and the number of DEGs was significantly higher than other light treatments. The numbers of extremely significant DEGs throughout fruit ripening were 544, 242 and 313, respectively, after RL, BL, and WL treatment (Fig. 3b). Notably, at the

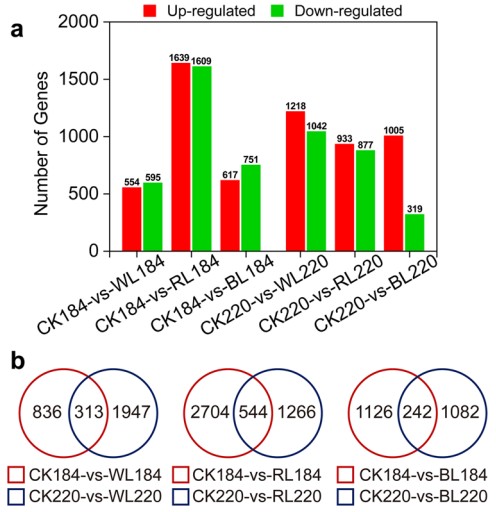

**Fig. 3 Profiling the changed genes and DEGs between grapefruit at different ripening stages. a** The histogram presents the number of upregulated and downregulated genes between samples during grapefruit ripening. **b** Venn diagram for DEGs between grapefruit samples at two ripening stages. "A" is the control group and "B" was the treatment group in "A-vs-B". CK: control group; WL: white-light treatment; RL: red-light treatment; BL: blue-light treatment. Red circle and blue circle represent 184 and 220 days after blossom, respectively.

two indicated detection points, the unique DEGs of RL, BL and WL treatment, respectively, reached a maximum of 2704 at 184 DAB, 1126 DEGs at 184 DAB and 1947 DEGs at 220 DAB.

**Weighted gene co-expression network analysis (WGCNA).** The WGCNA was performed using 12801 unigenes (FPKM > 1, the top 75% of variance), which were classified into twelve modules (Fig. 4, Supplementary Data 4), of which the 'blue' and 'turquoise' modules were remarkably associated with carotenoid metabolism under different light condition during fruit ripening ($p < 0.05$). The analysis of the correlation between gene expression and carotenoid accumulation demonstrated that the 'blue' module contained 4832 (37.7%) genes and was significantly positively correlated with the content of carotenoid with a correlation coefficient of 0.69 ($p = 0.0002$). However, 'turquoise' modules were highly negatively related with carotenoid content, which contained 6559 (51.2%), with a correlation coefficient of −0.68 ($p = 0.0003$). Interestingly, both 'blue' and 'turquoise' modules weren't associated with 9-*cis*-violaxanthin content ($p > 0.05$). These results indicate that genes in these modules were potentially correlated with carotenoid accumulation under different light conditions.

**Expression analysis of genes related to carotenoid metabolism.** In the carotenoid metabolic pathway (Fig. 5a), a total of eight structural genes, including *ZEP6*, *CCD4-1*, *NCED3* were differentially expressed in response to RL during ripening (Supplementary Table 5). RL significantly induced the transcription for carotenoid biosynthetic (*GGPPS2*, *PDS*, *Z-ISO*, *ZDS2/7* and *CHYB*) and down-regulated cleavage genes (*ZEP2*, *NXS* and *CCD1-2*) during grapefruit ripening (Fig. 5a, Supplementary Data 5). On the contrary, BL suppressed the expression of upstream genes in the carotenoid metabolic pathway. In WL-treated grapefruit, we found carotenoid cleavage genes were also upregulated to some extent.

Among the 11367 genes in the carotenoid co-expression modules, a total of 641 transcription factors (TFs), which were enriched in 65 gene families such as MYB (68), bHLH (41), ERF (39), NAC (29) WRKY (17) and MADS (16), were identified (Supplementary Figure 2). In order to further excavate the transcription factors responding to RL, we screened the differentially expressed transcription factors between RL and DS group. Subsequently, a total of forty-three TFs differentially expressed in response to RL, including ERF (7), NAC (6), bHLH (5), WRKY (4), SBP (3), MYB (2), MADS (2), HSP (2), C2H2 (2), bZIP (1), Dof (1) and GRAS (1) were identified as candidate TFs modulating carotenoid biosynthesis during fruit ripening (Fig. 5b, Supplementary Data 6). Among these genes, ERF023/062, NAC2-like/72, WRKY20, bHLH128, AGL61, MYB-like and SBP-like7/13 were down-regulated, while NAC21 and AGL11 were upregulated in response to RL during grapefruit ripening.

**Visualization of gene networks.** In order to identify the hub gene underlying carotenoid metabolism under different light-transmittance conditions, the co-expression for structural genes and regulators were visualized using Cytoscape. In 'blue' modules, eleven TF members—derived from the NAC (3), bHLH (2), bZIP (2), ERF (1), MYB (1), MADS (1) and HSP (1) families—were identified as the key genes related to carotenoid metabolism. Meanwhile, eighteen structural genes, namely *GGPPS1*, *PSY*, *PDS*, *Z-ISO*, *ZDS4/ZDS7*, *CRTISO2/3*, *CYP97C1/2*, *ZEP5/6*, *CCD1-1*, *VDE1/2*, *CCS*, *NXS*, *NCED1*, which directly involved in the carotenoid biosynthesis and cleavage, were identified as the key regulatory genes in co-expression network (Fig. 6a). In addition, we found that twenty-two TFs, including ERF (6), WRKY (5), SBP (4), NAC (3), bHLH (2), MYB (1) and MADS (1) family members, were co-expressed with carotenoid upstream genes (*GGPPS2* and *ZDS2/3/5*) and cleavage gene (*CYP97A*, *ZEP2* and *CCD1-2*) in 'turquoise' modules (Fig. 6b). Notably, there were a

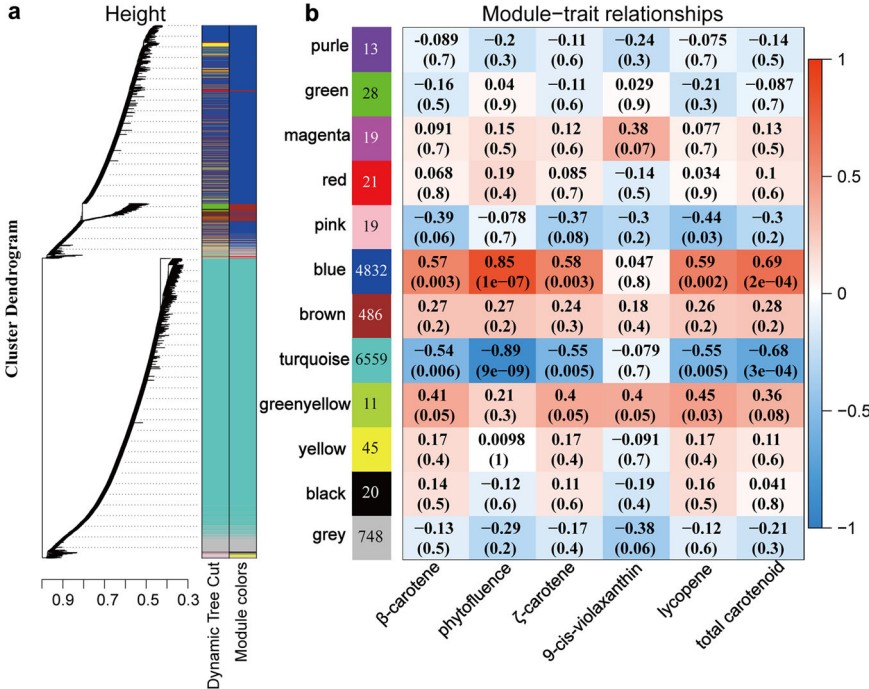

**Fig. 4 Weighted gene co-expression network analysis of grapefruit during ripening under different light-transmittance treatments. a** Hierarchical clustering tree displays twelve modules of co-expressed genes, in which each leaf represents one gene. **b** Modules related to carotenoid and corresponding *p*-values. The left panel indicates twelve modules and the number of genes contained by each module. The right panel displays a color scale for module and trait correlations from −1 to 1.

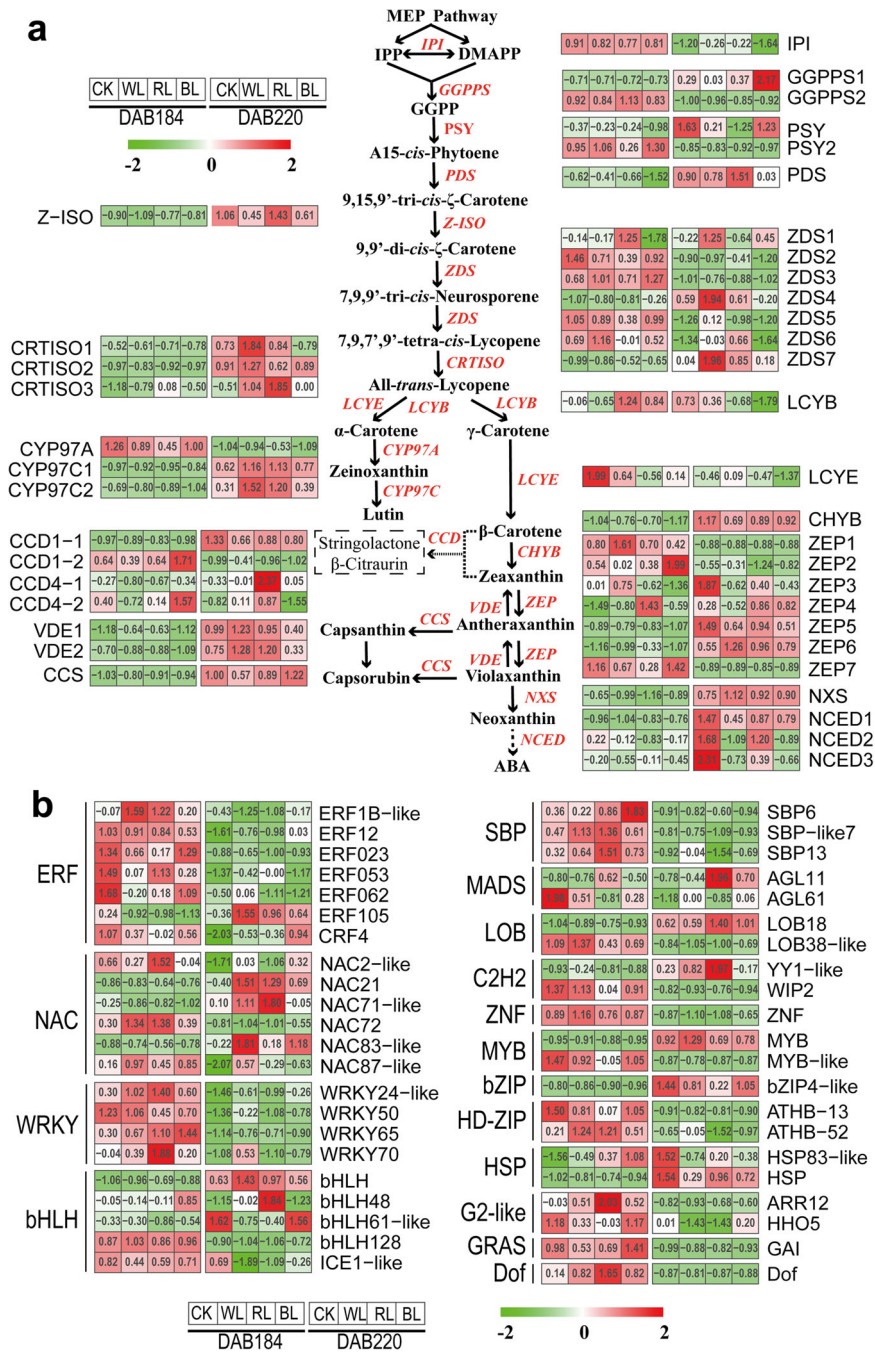

**Fig. 5 Heatmap analysis of genes correlated with carotenoid metabolism during grapefruit ripening. a** The heatmap analysis of structural genes correlated with carotenoid metabolism during grapefruit ripening. **b** The heatmap analysis of transcriptional factors correlated with carotenoid metabolism during grapefruit ripening. Row and column indicate gene names and samples in the heatmap, respectively. Red, white and green represent high, medium and low expression level for genes. CK: control group; WL: white-light treatment; RL: red-light treatment; BL: blue-light treatment.

co-expression relationship between *GGPPS2* and twenty-three TFs and *CYP97A* were co-expressed with twenty TFs (Supplementary Data 7 and Supplementary Data 8). These results suggested that above transcription factors and structural genes might interact with each other to regulate the flux for carotenoid in grapefruits.

## Discussion
Light signals play a vital role in carotenoid metabolism[23,27]. Although many studies have investigated the effects of

postharvest light treatments, such as LED, pulse, and ultraviolet light, on carotenoid metabolism[28–30]. As a usual agricultural practice in citrus cultivation, the role of light-transmittance bagging in carotenoid metabolism underlying grapefruit ripening previously had remained elucidated. In this work, different light-transmittance bagging treatments were used to understand the influence of light quality on grapefruits throughout the ripening process at the metabolic and molecular levels.

Existing research indicates light irradiation modulates the biosynthesis and catabolism of carotenoids in fruit and modifies the concentration and composition of carotenoids[27,31]. The total

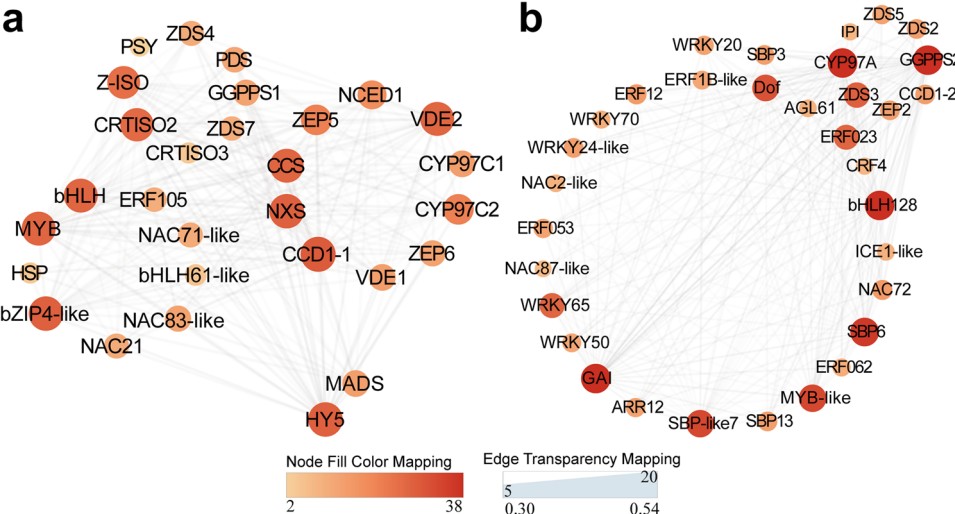

**Fig. 6 The co-expression network of TFs and structural genes related to carotenoid metabolism in co-expression module. a** The co-expression network of TFs and structural genes related to carotenoid metabolism in 'blue' module. **b** The co-expression network of TFs and structural genes related to carotenoid metabolism in 'turquoise' module (B)Modules related to carotenoid and corresponding p-values. Dot sizes and colors represent the numbers for correlated genes.

carotenoid content was reduced upon ripening in covered tomatoes and peppers[25]. In grapefruit peel, an unusual pattern of lycopene accumulation might be associated with the developmentally regulated differentiation of chromoplasts mediated by light[31]. Light deprivation promoted peel degreening and reduced carotenoid accumulation in mandarins and sweet orange fruits[27]. In contrast, light irradiation enhanced carotenoid accumulation and external quality during mandarin fruit development[32]. Here, our results showed that carotenoid (total carotenoid, β-carotene, ζ-carotene, lycopene and 9-*cis*-violaxanthin) accumulation was significantly induced by red light-transmittance bagging treatments (Fig. 2), suggesting that RL played significant positive role in carotenoid accumulation in grapefruit pulp. These are evidence of the promotion of carotenoid metabolism by light in fruit.

Lighting factors can regulate secondary metabolism by light quality, light intensity, and light irradiation time in plant[25]. Red LED light (660 nm) activated the expression of *VvNCED1* in ripening grape skin (*Vitis vinifera* L.)[33], but had no effect on the carotenoid content of citrus juice sacs. Blue LED light (470 nm) treatment stimulated carotenoid accumulation by upregulating the expression of the *CitPSY* gene in the juice sacs of three citrus varieties (mandarin (*Citrus unshiu* Marc.), Valencia orange (*C. sinensis* Osbeck) and Lisbon lemon (*C. limon* Burm.f.)[34]. In the present study, we found that RL promoted lycopene and β-carotene accumulation in grapefruit pulp along with upregulation of *GGPPS2*, *PDS*, *Z-ISO*, *ZDS2/7* and *CRTISO3* and downregulation of *CYP97A*, *ZEP2* and *CCD1-2* (Figs. 2 and 5), which was similar with the upregulation of the *PSY* gene induced by continuous red light in *Arabidopsis thaliana* seedlings, leading to increases in carotenoid content[35]. However, blue- and white-light transmittance treatment have no significant effect on carotenoid content during grapefruit ripening. These results showed that red light-transmittance bagging play important regulatory role in the carotenoid metabolism of grapefruit.

A large number of TFs have been reported to be involved in carotenoid metabolism via transcriptional regulation of key structural genes in plant. Here, multiple members of ERF, NAC, WRKY, MYB, MADS, bHLH families be identified as hub genes for modulating carotenoid flux (Figs. 5 and 6). Work in

*Arabidopsis thaliana* revealed that the PIF1 (phytochrome interacting factor 1) transcription factor suppressed *PSY* transcript by directly binding the G-box motifs and further regulated carotenoid metabolism[36]. Zhou et al. showed that CpbHLH1/2 promoted lycopene degradation to carotenoids by upregulating the transcripts of the lycopene β-cyclase genes (*CpLCYB* and *CpCHYB*) in response to strong light during papaya ripening[19]. Blue- and red-light supplementation irradiation to tomato fruits at anthesis facilitated lycopene biosynthesis, which was considered to be related with regulation of the photoreceptor HY5 (ELONGATED HYPOCOTYL5) and PIFs upon the expression of *PSY1*[37]. Here, in RL-treated grapefruit, bHLH128 in 'turquiose' module negatively correlated with lycopene presented markedly down-regulated trend coupled with transcript increase of *GGPPS2*, *PDS* and *Z-ISO* (Figs. 2, 5, 6 and 8), which accounted for higher lycopene level, and these results are paralleled with the carotenoid increase of content in SlPIF4-silenced tomato[38].

Some ripening related regulators have been shown the regulatory role in carotenoid metabolism. In tomato, MADS-box TFs RIN gene was reported to specifically regulating the accumulation of lycopene by positively regulated carotenoid biosynthetic genes (including *PSY*, *Z-ISO*, *CRTISO*) and negatively regulated carotenoid downstream genes *LCYB* and *LCYE*, while FUL homologs FUL1/2 regulated overall carotenoid pathway by targeted multiple carotenogenic genes[39,40]. In sweet orange (*Citrus sinensis*), CsMADS5/6 activated expression for carotenogenic genes, including *PSY*, *PDS*, *LCYb1/CCD1* via directly binding its promoter and thus modulated carotenoid metabolism[14,15]. In 'blue' module, we found that up-regulation of AGL11 in response to RL was accompanied by the increased expression level of *PDS*, *Z-ISO*, *CRTISO3*, which facilitated carotenoid accumulation (Figs. 2, 5 and 8). By contrast, the significantly reduced expression of AGL61 for 'turquoise' module during ripening suggested their negative correlation with carotenoid accumulation in grapefruit (Figs. 5 and 8). Recently, PpERF3 has been shown to be involved in ABA biosynthesis by activating *PpNCED2/3* transcription during peach fruit ripening[41]. Here, PpERF3 homolog ERF023 was down-regulated in response to RL, which suggested ERF023 were highly likely to be involved in carotenoid metabolic process

mediated by RL (Fig. 5 and Supplementary Figure 3). On the contrary, the transcript of ERF105 was remarkably promoted by RL treatment and NCED1 shared similar expression patterns with ERF105 (Fig. 5 and Fig. 8). The above analysis indicated ERF TFs differently respond to RL and collaboratively regulated carotenoid accumulation. In the Arabidopsis, suppression of AtRAP2.2 leads to a reduction of PSY and PDS transcript[42]. In rice leaves, AP2/ERF genes were negatively associated with carotenoid accumulation under both blue- and red-light treatments[43]. Here, we also found multiple ERF TFs (ERF1B-like/12/053) in 'turquoise' module were downregulated and displayed negative correlation with carotenoid content in response to RL.

Another fruit ripening related TF NACs were also reported to be involved in carotenoid metabolism. In tomato SlNAC4/19/48 RNAi fruit, the transcript levels of PSY were reduced and thus resulted in decreased lycopene[44,45]. However, the overexpression of SlNAC1 reduced lycopene content, which was associated with a reduction in SlPSY and an increase in SlLCYB and SlLCYE expression[46]. During papaya fruit ripening, CpNAC2 co-operated with CpEIN3a to promote CpPDS2/4, CpZDS, CpCHYB, and CpLCYE transcription, accounting for the elevated carotenoid contents[18]. CcNAC1/2 were transcriptionally upregulated under red-light treatment in Citrullus colocynthis[47]. Similarly, FcrNAC22 upregulated carotenoid metabolism and ABA synthesis via activation of FcrLCYB1, FcrBCH2 and FcrNCED5 in RL-irradiated fruits[13]. Here, we observed that the increased expression levels of NAC21 (the FcrNAC22 homolog) mediated by RL were positively related with expression of upstream genes (PDS, Z-ISO, ZDS2/7 and CHYB) in the carotenoid metabolic pathway, while downregulated NAC2-like (SlNAC1 homolog) /NAC72 showed a negative correlation with transcript for these genes, consistent with lycopene accumulation in ripening grapefruit fruit (Figs. 2, 5 and 8; Supplementary Figure 3).

Some publications in recent years have reported that MYB TFs played a positive role in carotenoid regulation. In the flavedo of Citrus reticulate, CrMYB68 indirectly inhibited the transformation of α/β-carotene via negative regulation for CrBCH2 and CrNCED5[16]. AdMYB7 was positively correlated with AdLCYB in terms of expression and further regulated carotenoid biosynthesis[21]. In 'turquoise' module, we found that MYB-like inhibited by RL was also negatively correlated with carotenoid accumulation, especially β-carotene, during grapefruit ripening (Figs. 4 and 5). Additionally, two WRKY TFs, namely WRKY50/70 were differentially expressed in response to RL during grapefruit ripening (Fig. 5). In Osmanthus fragrans, OfWRKY3 was found to be a positive regulator of the OfCCD4 gene via binding to its W-box palindrome motif[48]. In this study, we also observed that WRKY20, the homolog of OfWRKY3, were gradually downregulated as grapefruit fruit ripening, accompanied by the reduction of CCD1-2 expression of in RL treatment (Figs. 2, 5 and 8). Besides, RL also notably suppressed expression for SBP-like7/13 and Dof TFs in 'turquoise' module, suggested these TFs might involve in carotenoid accumulation (Fig. 5).

In conclusion, the carotenoid accumulation in grapefruit responds differently to light-transmittance bagging, RL have the significant inducing role during fruit ripening. The process was modulated by multiple TFs (bHLH128, NAC2-like/21/72, MYB-like, AGL11/AGL61, ERF023/062, WRKY20, SBPlike-7/13) as well as carotenogenic genes (GGPPS2, PDS, Z-ISO, ZDS2/7 CHYB and CCD1-2). Based on verification by qPCR (Fig. 7, Supplementary Data 9), a preliminary regulatory model of red light-transmittance bagging-induced carotenoid metabolism in grapefruits was established (Fig. 8). These findings not only provide new insight into the regulation of carotenoid metabolism, but also offer an effective approach for enhancing the quality of citrus fruits in agricultural practice.

## Methods

**Plant materials and treatments**. 'Huoyan' grapefruit were cultivated at the National Citrus Germplasm Repository of the Citrus Research Institute at the Chinese Academy of Agricultural Sciences in Chongqing, China and used as experimental materials. Trees with the same age, tree structure, and identical growth conditions were selected for the experiment and cultivated under the same management condition. Grapefruits with similar sizes and colors from outside of the tree were bagged with four different light-transmitting paper bags at 120 days after blossom (DAB)—red-light-transmitting bags (RL) (peak wavelength, 748 nm), blue-light-transmitting bags (BL) (peak wavelength, 478 nm), white-light-transmitting bags (WL) and no bagging (CK) was as the control group (Supplementary Figure 4). There is no difference in permeability of the different light-transmitting bags for oxygen and $CO_2$ between difference color bags, and also no significant difference for other micro-climate parameters were observed between different light-transmitting bags (Supplementary Figure 4b and Supplementary Figure 4c). Fruits of a uniform size were picked at 184 (maturation) and 220 (fully ripe) days after blossom (DAB). Each fifteen fruits were as one replicate and three biological replicates were used for each sample point of every treatment. After determining the basic physiological parameters, the editable juice vesicles of grapefruits were cut into small cubes, frozen using liquid nitrogen, and stored at −80 °C for further analysis.

**Determination of basic physiological parameters**. The fruit color parameters were measured using the Hunter Associates Laboratory Scanner (Hunter Associates Laboratory, Inc., Reston, VA, USA). The citrus color index (CCI) for the mesocarp was calculated according to the formula $CCI = 1000 \times a^* / (L^* \times b^*)$, using five fruits as a single replicate and three biological replicates were used for each sample. To determine the total soluble solid (TSS) content, 200 μL of fresh-squeezed juice was obtained from juice sacs and then analyzed with a digital hand-held refractometer (Atago PR-101R, Atago, Japan). Titratable acidity (TA) was measured after the juice sample was diluted 50 times with purified water.

**Extraction and identification of carotenoids**. Carotenoids were identified following our constructed method[49]. Ten grams of pulp powder was extracted with 20 mL of solvent (hexane/acetone/ethanol, 50:25:25, v/v/v) in a screw-top tube. The colored top layer was recovered and dried with nitrogen gas after being left to stand for 30 min, protected from light. After saponification, 2 mL of 1% butylhydroxytoluene (BHT)/methyl tert-butyl ether (MTBE) was added to the colored layer, and the mixture was filtered through sodium sulfate into a brown bottle for drying. The residue was dissolved in 2 mL of methanol/acetone (2:1, v/v) for HPLC analysis.

The carotenoids were identified by HPLC (Waters, Milford, MA, USA) with a $C_{30}$ chromatography column (250 × 4.6 mm, 5 μm; YMC, Wilmington, NC, USA). The mobile phases for the carotenoids were composed of MTBE (A), methanol (B), and an aqueous phase (C) and were prepared by a multistep linear gradient elution. The identification was performed by comparing the retention times and UV–visible spectral peaks between the samples and standards. The carotenoid contents were calculated according to a standard curve based on authentic compounds and are expressed herein as mg/kg fresh weight (FW).

**Library construction, transcriptome sequencing, and gene annotation**. Total RNA was extracted using an Agilent RNA 6000 Nano kit (Agilent, CA, USA) according to the manufacturer's instructions, the RNA concentration and integrity were assessed using an Agilent 2100 Bioanalyzer, and the OD260/OD280 and OD260/OD230 values were determined using a NanoDrop 2000 spectrophotometer (NanoDrop 2000, Wilmington, DC, USA) to assess the RNA purity. Twenty-four mRNA libraries were constructed for RNA-seq of the mesocarp samples harvested at 184 and 220 DAB. Three biological replicates were performed for each sample.

RNA-Seq libraries for maturation and fully ripe-stage grapefruit under CK, RL, BL, and WL treatment were constructed using TruePrepTM DNA Library Prep Kit V2 for Illumina® (Vazyme, Nanjing, China) according to manufacturer's manual. The libraries were sequenced on the MGISEQ-2000 system at the Beijing Genomics Institute (BGI), China. The raw sequencing data were filtered by removing adaptors, low-quality and redundant sequences, and reads with unknown "N" base content higher than 5% using the SOAPnuke (version 1.4.0) and Trimmomatic (version 0.36) software. The clean reads were aligned to the reference genome database (accession number: AJPS00000000) using HISAT (version 2.1.0)[50].

For transcription factor annotation, open reading frames (ORF) were obtained from the quality-checked data using getorf (EMBOSS: 6.5.7.0, http://emboss.sourceforge.net/apps/cvs/emboss/apps/getorf.html, -minsize 150) and aligned to Plant Transcription Factor Database (http://planttfdb.gao-lab.org/blast.php).

**Identification of differentially expressed genes (DEGs)**. The RSEM software package (version 1.2.8, http://deweylab.biostat.wisc.edu/rsem/rsem-calculate-expression.html) was used to calculated expression levels for transcripts with the default parameters[51]. The expression levels are expressed as FPKM values. The genes that were differentially expressed between two samples were determined

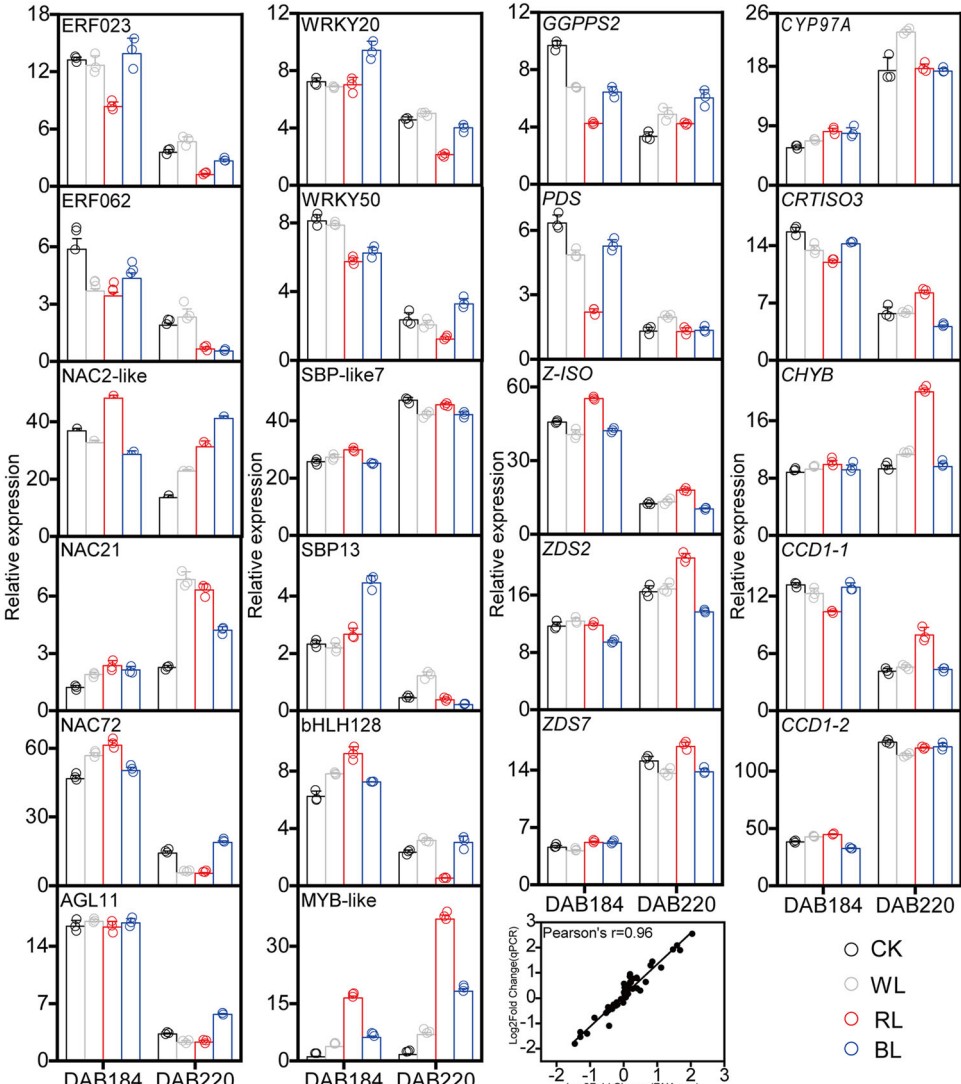

**Fig. 7 Expression confirmation of 22 important candidate genes involved in the carotenoid metabolism using qRT-PCR and Coefficient analysis between the RNA-seq FPKM and qRT-PCR values.** The relative expression of target genes relative to a control gene is shown with standard errors. Three biological replicates were used for each point. CK: control group; WL: white-light treatment; RL: red-light treatment; BL: blue-light treatment. FC: fold change between WL, RL, BL and DS. The circle represents the individual data point derived from three independent experiments.

according to the Poisson distribution and the algorithm developed by BGI[52]. The DEGs ($p \le 0.05$; |log$_2$ fold change| ≥1) were then screened for further analysis.

**Weighted gene co-expression network analysis (WGCNA) and network visualization for candidate genes.** A total of 17070 unigenes with FPKM values > 1 were utilized to conduct weighted gene co-expression network analysis using WGCNA[53], reshape2 and stringr packages in Rstudio (v1.4.1717, https://www.rstudio.com/products/rstudio/download/). To reduce the size of the data calculation, a total of 12801 unigenes with the first 75% were screened from the above unigenes with unsigned TOM type to build a co-expression network. The phenotypic data regarding the carotenoids in the pulp were associated with the constructed co-expression network to screen the modules that were significantly correlated with carotenoid metabolism ($p \le 0.05$). Finally, DEGs were imported into the Cytoscape software (version 3.7.2, https://cytoscape.org/download.html) for network visualization.

**Phylogenetic analysis.** Protein sequence alignment between candidate transcription factors and transcription factors (Supplementary Table 6) related to carotenoid metabolism that had been reported were performed by ClustalW program in MEGA7. Based on the results of sequence alignment, the phylogenetic tree was established by neighbor-joining method with 1000 bootstrap replicates.

**Real-time quantitative PCR.** The expression levels of candidate structural genes and TFs were verified by qRT-PCR analysis. Actin gene (*Citrus sinensis* actin-7: LOC102577980) expression was used as a normalization reference. Specific primers were designed using Primer5 (Supplementary Data 10). Gene expression levels were detected using an iQ5 instrument (Bio-Rad Laboratories, Inc. America) with the SYBR® Premix Ex TaqTM II Kit (TaKaRa Biotechnology (Dalian) Co, Ltd, China). The amplification program was as follows: 95 °C for 1 min, followed by 40 cycles at 95 °C for 20 s, 58 °C for 20 s and 72 °C for 30 s. Melting curve analysis was done to ensure the specificity for each amplification product. $2^{-\Delta\Delta CT}$ was used to calculate the relative expression level of genes[54]. Each qRT-PCR analysis was performed in triplicate and the mean value was used for the qRT-PCR analysis.

**Statistics and reproducibility.** For basic physiological parameters (including TSS, TA and CCI), HPLC analysis for carotenoid, transcriptome sequencing and quantitative PCR verification, three biological replicates were performed. Presented data were derived from the mean of three independent experiments ± the standard deviation. In all bar graphs, Individual data points are shown alongside the mean and standard deviation. Statistical difference analysis was conducted via using IBM SPSS Statistics software. Pairwise comparisons between means were carried using Student Newman Keuls test at the significance level $P < 0.05$. The correlation analysis was completed by Origin 2018.

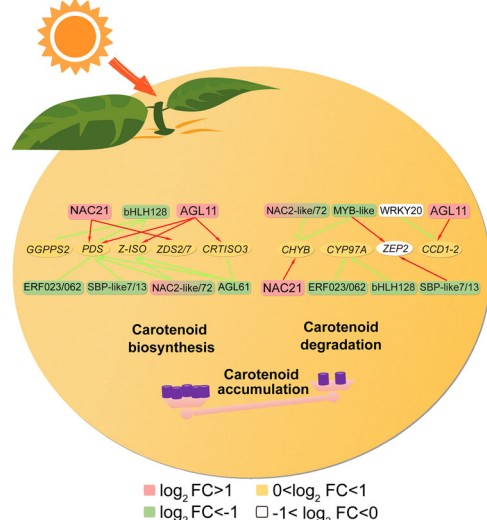

**Fig. 8 The proposed models of carotenoid metabolism mediated by red light during grapefruit ripening.** Pale pink and light green rounded rectangle, respectively, represents up- and down-regulated TFs in response to RL. Red and green arrow indicate positive and negative regulation of TFs on structural genes, respectively. Oval suggests structural genes in carotenoid metabolic pathway. Different colors represent up- and downregulation level for genes in response to RL. FC: fold change between RL and DS; MYB, MYB transcription factor; NAC, NAC transcription factor; MADS, MADS-box transcription factor; ERF, Ethylene response factor; AGL, amylo-alpha-1,6-glucisidase, 4-alpha-glucanotransferase; bHLH, basic helix-loop-helix transcription factor; WRKY, WRKY transcription factor; *GGPPS2*, geranylgeranyl diphosphate 2; *PDS*, phytoene desaturase; *Z-ISO*, ζ-carotene isomerase; *ZDS*, ζ-carotene desaturase; *CRTISO*, carotenoid isomerase; *CYP97A*, β-ring hydroxylase; *CHYB*, β-carotene hydroxylase; *ZEP*, zeaxanthin epoxidase; *CCD*, carotenoid cleavage dioxygenase.

**Reporting summary**. Further information on research design is available in the Nature Research Reporting Summary linked to this article.

## Data availability

The transcriptome raw reads have been deposited to BioProject (https://submit.ncbi.nlm.nih.gov/subs/sra/) under accession number PRJNA796840. Source data underlying figures in the main text are presented in Supplementary Data 1-10. All other data in present study are available from the corresponding author on reasonable request.

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

## Acknowledgements
This work was supported by the National Science Foundation of Chongqing for Distinguished Young Scholars (No. cstc2019jcyjjq0029).

## Author contributions
W.P.X. designed the project. X.L.H. prepared the manuscript. X.L.H. and L.P.H. analyzed the data. C.Y., L.P.H. and X.L.H. participated in assaying the physiological parameters. W.B.K. participated in collecting the materials.

## Competing interests
The authors declare no competing interests.
