## [Peer Review File · Communications Biology]

Reviewers' comments:

Reviewer #1 (Remarks to the Author):

This manuscript study the effects of different light-transmitting bags on carotenoid metabolism and transcriptome during fruit ripening of grapefruit, and found that red-light-transmitting bags treatments have great potential to improve coloration and nutritional quality of citrus fruits. It is benefit to the bagging in production of citrus fruits.

There are some comments as follow:

Line 108-110 "the majority of studies have focused on the effect of postharvest light treatment on fruit quality, with only a few referring to the impact of developmental light treatment on fruit quality ". There are some paper about light effects on fruit quality. The following paper may be the newest one about red light affecting citrus fruit carotenoid.

Jinli Gong, Yunliu Zeng, Qiunan Meng, Yajie Guan, Chengyang Li, Hongbin Yang, Yingzi Zhang, Charles Ampomah-Dwamena, Ping Liu, Chuanwu Chen, Xiuxin Deng, , Yunjiang Cheng and Pengwei Wang. Red light-induced kumquat fruit coloration is attributable to increased carotenoid metabolism regulated by FcrNAC22. Journal of Experimental Botany, <https://doi.org/10.1093/jxb/erab283>

Line 105-106 "light ----- as the crucial medium for the exchange of information between the plant and external environment" is this exactly explain?

Line 154 "ζ-carotene lycopene "is "ζ-carotene, lycopene "?

Line 226 "metabolism underlying fruit ripening previously had not been elucidated " There are some paper about light effects on fruit quality during development.

Line 350-352 "red-light-transmitting bags (RL) (peak wavelength, 748 nm), blue-light-transmitting bags (BL) (peak wavelength, 478 nm), and white- light-transmitting bags (WL) ".

From Fig. S1 The transmittance of far-red (peak wavelength, 748 nm) are similar between RL and WL. Due to higher transmittance from 450-748 nm in WL, the effects of bagging treatments on carotenoid in grapefruit may be intergrated effects of light intensity and quality. So the transmitted light spectrum under different bags at same light intensity of sun light source should measured.

Reviewer #2 (Remarks to the Author):

This is an interesting manuscript, however, I feel validation is required of the major conclusions. I appreciate that expression of these gene correlates with enhanced carotenoids content. However, this does not prove that their expression is causal. This would require some form of genetic and or pharmacological experiment targeting the genes and gene products respectively. In the absence of such validation the main conclusion of the manuscript is not sufficiently supported. Besides this major flaw the manuscript is well presented and contains generally well carried out experiments however although the most parsimonious conclusion it definitely requires further support.

Reviewer #3 (Remarks to the Author):

Authors of the manuscript no COMMSBIO-21-2202 have performed the transcriptomic and metabolic analysis of 'Huoyan' grapefruit subjected to different light signals during fruit ripening. The aim of the study was to elucidate the role of light quality on the carotenoid accumulation in grapefruit and to understand the transcriptional regulatory mechanism underlying light signals during fruit ripening. The authors showed that the carotenoid accumulation in grapefruit responded differently to light quality, and that the red light have significantly inducing role during fruit ripening. The process was modulated by multiple TFs as well as carotenogenic genes. Based on detected DEGs, carotenoids determination and WGCNA, a preliminary regulatory model of red light-induced carotenoid metabolism in grapefruits was proposed.

Regarding the experimental part of work, the authors have used advanced and suitable methodology. The experiments are rather well designed, carefully performed and seem quite

reliable (questionable issues listed below). The information included in tables and figures is clear (apart from issues commented below). The results are appropriately discussed; the novelty of presented work is justified; hypotheses are drawn and the potential for the industrial application is highlighted. Except for minor stylistic and typing mistakes, the manuscript is written with adequate English. The work has a potential to bring novel information to the field of study, however, in its current form it needs a major revision. There are several major and minor issues which should be addressed or explained before considering this work for publishing in Communications Biology.

Major points:

1. Materials and methods; Plant material and treatments section: please, add the information about the permeability of the bags for oxygen and CO₂.
2. Materials and methods; Plant material and treatments section: In line 356 you say that "fruits were cut into small cubes". Please, explain exactly, which part (tissue) of the grapefruit was used for each particular analysis. Use botanical terms.
3. Materials and methods; Plant material and treatments section: You have used dark-shading bags (DS) as control conditions in your experiment. Is there any particular reason why a daylight was not included as a control?
4. Materials and methods; Library construction section: Have you estimated the RNA integrity number (RIN) for your RNA samples and was it high enough for the library construction and RNAseq? What was the name of the Illumina kit for the library synthesis/ oligo (dT) beads/ adapters - add names of the products and names of manufacturers.
5. Materials and methods; Library construction section: There is no record of the PRJNA728380 BioProject in the SRA (NCBI) database. You state that eight mRNA libraries were constructed for RNA-seq (4 light treatments x 2 harvest points), each in three biological replicates. That would make 24 libraries sequenced in total. Table S2 and Table S3 does not confirm that and the absence of the BioProject from the SRA database does not allow to verify the information of the number of biological replicates sequenced. Please, comment on that and make the SRA information available to the reviewers and readers.
6. Results; Expression analysis of genes related to carotenoid metabolism section: Please, elaborate more on results. Did only red-light influenced gene transcription significantly? How about other treatments (Figure 5A)? Lines 196-199: the same sentence used twice.
7. Figure 1 and Figure 2: Provide the ranges of values for the statistical difference calculated with one-way ANNOVA. Explain between which samples was comparison made in each case? In other words – what the letters a – f mean?
8. Is there any particular reason, why the gene expression results from RNA-seq were not confirmed by qRT-PCR for at least some of the target genes?

Minor points:

1. Line 122: it should be "based on"
2. Line 156: were the differences significant?
3. Line 159: what does mean "for simples"?
4. Line 234: Delete the word "in"
5. Line 250: In which tissue?
6. Line 320: should be: "positively correlated"

7. Line 332: should be: "responds"

8. Figure 3: colours of the Venn diagrams are slightly misleading. Try to use stick to the colours of your light treatments and use red, blue, black and grey (for white), or – do not use colours at all.

9. Figure 7. Explain all abbreviations used in the figure in the figure description.

10. The Table S7 is not cited in the main text.

11. Provide the reference number of the grapefruit reference genome which was used for the gene annotation here.

Summarizing, I would recommend presented manuscript for reconsidering after correcting the major and minor issues listed above.

Reviewer #1:

Question 1: Line 108-110 'the majority of studies have focused on the effect of postharvest light treatment on fruit quality, with only a few referring to the impact of developmental light treatment on fruit quality'. There are some paper about light effects on fruit quality. The following paper may be the newest one about red light affecting citrus fruit carotenoid. Jinli Gong, Yunliu Zeng, Qiunan Meng, Yajie Guan, Chengyang Li, Hongbin Yang, Yingzi Zhang, Charles Ampomah-Dwamena, Ping Liu, Chuanwu Chen, Xiuxin Deng, Yunjiang Cheng and Pengwei Wang. Red light-induced kumquat fruit coloration is attributable to increased carotenoid metabolism regulated by FcrNAC22 (2021).

Answer: We accepted your suggestion. the reference was added in the revised manuscript, see reference list No.13 in line 514-516, and the description was modified as “only several works referring to the impact of developmental light treatment on fruit quality” in line 110.

Question 2: Line 105-106 “light-as the crucial medium for the exchange of information between the plant and external environment is this exactly explain?”

Answer: We are sorry for our unclear description. The sentence was corrected as “as an environmental signal to participate in the regulation of a variety of metabolic processes during plant development”. See lines 105-107.

Question 3: Line 154 “ζ-carotene lycopene ”is “ζ-carotene, lycopene”?

Answer: Yes, it should be “ ζ -carotene, lycopene”, it was corrected, see line 146-147.

Question 4: Line 226 "metabolism underlying fruit ripening previously had not been elucidated" There are some paper about light effects on fruit quality during development.

Answer: We accepted your suggestion. We modified the sentence as " As a usual agricultural practice in citrus cultivation, the role of light-transmittance bagging in

carotenoid metabolism underlying grapefruit ripening previously had remained elucidated." and as is shown in line 234-236.

Question 5: Line 350-352 red-light-transmitting bags (RL) (peak wavelength, 748 nm), blue-light- transmitting bags (BL) (peak wavelength, 478 nm), and white-light-transmitting bags (WL) From Fig. S1 The transmittance of far-red (peak wavelength, 748 nm) are similar between RL and WL. Due to higher transmittance from 450-748 nm in WL, the effects of bagging treatments on carotenoid in grapefruit may be integrated effects of light intensity and quality. So, the transmitted light spectrum under different bags at same light intensity of sun light source should be measured.

Answer: Thank you for pointing it out.

Sure, the same intensities of the light irradiation should be used in the similar study, which seems to be convenient to be compared. In doing so, if the intensity is the same under different light-transmitting bagging, the effect of light on carotenoid will only light quality.

As an effective method of protecting fruit from insect infestations, bird attack, and sunburn as well as reducing disease incidence rate and chemical residues, fruit bagging is extensively used in modern orchards. And the different light-transmitting bags used in the study is the bags widely used in citrus cultivation in China.

However, the protection cultivation and LED should be used for excise control the light intensity of each light quality, the cost is high, it is unbenefited for extension to practice. So, in the study, we only compared the regulation effect of these bags widely used in practice on fruit quality. Our results showed that red-light-transmitting bags (RL) is the optimal bags. It will be used in practice.

According to your comment, the light intensity was completed in Supporting information Fig.1C. In addition, other micro-climate parameters of different light-transmitting bags were also added in the Supporting information Fig.1C. Yes, you are right, the intensity under different light-transmitting bags are different. So the effects of bagging treatments on carotenoid in grapefruit is integrated effects of light intensity and quality. Based on it, the manuscript title was modified as "Transcriptomic and metabolic analysis uncovers the role of red light-transmittance bagging in promoting carotenoid accumulation of grapefruit during ripening".

Furthermore, the related discussion was also modified, see lines 364-366.

Reviewer #2 (Remarks to the Author):

This is an interesting manuscript; however, I feel validation is required of the major conclusions. I appreciate that expression of these gene correlates with enhanced carotenoids content. However, this does not prove that their expression is causal. This would require some form of genetic and or pharmacological experiment targeting the genes and gene products respectively. In the absence of such validation the main conclusion of the manuscript is not sufficiently supported. Besides this major flaw the manuscript is well presented and contains generally well carried out experiments however although the most parsimonious conclusion it definitely requires further support.

Answer: Thank you for your comments. The expression of these gene related with carotenoids content and some regulators was validated by qPCR. See the Fig. 7 in line 689-694 in revised manuscript.

Reviewer #3:

Question 1: Materials and methods; Plant material and treatments section: please, add the information about the permeability of the bags for oxygen and CO₂.

Answer: We accepted the suggestion. the information about the permeability of the bags for oxygen and CO₂ was added in the Supporting information Fig.S1C, and related description was also complemented in the revised manuscript, see lines 364-366.

Question 2: Materials and methods; Plant material and treatments section: In line 356 you say that 'fruits were cut into small cubes'. Please, explain exactly, which part (tissue) of the grapefruit was used for each particular analysis. Use botanical terms.

Answer: We are very sorry for our unclear descriptions. In fact, only the edible juice vesicles of grapefruits were used for analysis. The related information was added in 370.

Question 3: Materials and methods; Plant material and treatments section: You have used dark-shading bags (DS) as control conditions in your experiment. Is there any particular reason why a daylight was not included as a control?

Answer: Thank you your comments.

In 2020 we used dark as the control. To verify the treatment effect, in 2021 we redo the experiment using natural light (no bagging). As doing so, all 2021 data were used in the revised manuscript. For ensure the consistency the results, all transcriptomic and metabolic data was analyzed using 2021 data. And all graphs were made based on the new analyzed results. See the revised manuscript.

Question 4: Materials and methods; Library construction section: Have you estimated the RNA integrity number (RIN) for your RNA samples and was it high enough for the library construction and RNA-seq? What was the name of the Illumina kit for the library synthesis/ oligo (dT) beads/ adapters - add names of the products and names of manufacturers.

Answer: Thank you for point this out.

We evaluated the RNA integrity number (RIN) for all RNA samples and RIN score ranged from 7.8~9.2, which suggested RNA integrity is qualified (Supplementary Table S2). Therefore, sample can be used for RNA-seq.

RNA-Seq libraries for maturation and fully ripe-stage grapefruit under CK, RL, BL and WL treatment were constructed using TruePrep™ DNA Library Prep Kit V2 for Illumina® (Vazyme, Nanjing, China) according to manufacturer's manual and seen in line 410-412.

Question 5: Materials and methods; Library construction section: There is no record of the PRJNA728380 BioProject in the SRA (NCBI) database. You state that eight mRNA libraries were constructed for RNA-seq (4 light treatments x 2 harvest points), each in three biological replicates. That would make 24 libraries sequenced in total.

Table S2 and Table S3 does not confirm that and the absence of the BioProject from the SRA database does not allow to verify the information of the number of biological replicates sequenced. Please, comment on that and make the SRA information available to the reviewers and readers.

Answer: Thank you for point this out. Yes, three replicates for each sample were used in the study and thus there were twenty-four mRNA libraries. All of the obtained sequences from grapefruit were deposited in the NCBI Sequence Read Archive (SRA) repository under accession number PRJNA796840 and was released. in total, and seen in line 469-470.

Question 6: Results; Expression analysis of genes related to carotenoid metabolism section: Please, elaborate more on results. Did only red-light influenced gene transcription significantly? How about other treatments (Figure 5A)? Lines 196-199: the same sentence used twice.

Answer: We accepted your advice. They were elaborated as following: On the contrary, BL suppressed expression of up- and down-stream genes in carotenoid metabolic pathway. In WL-treated grapefruit, we found carotenoid pathway genes were also upregulated to some extent. The related description was added, seen line 196-199. In addition, we have deleted repeated sentence in our revised manuscript.

Question 7: Figure 1 and Figure 2: Provide the ranges of values for the statistical difference calculated with one-way ANNOVA. Explain between which samples was comparison made in each case? In other words - what the letters a - f mean?

Answer: Thank you for your comments. The values for the statistical difference calculated with one-way ANNOVA was added in revised manuscript, see lines 639 and 646. Different letters (a-h) indicate statistically significant difference in one-way ANOVA analysis and we had explained meaning of a-h in the corresponding figure legend.

Question 8: Is there any particular reason, why the gene expression results from RNA-seq were not confirmed by qRT-PCR for at least some of the target genes?

Answer: Based on your comments, a total of 22 candidate genes from heatmap in carotenoid metabolism pathway and co-expression network were confirmed by PCR.

As is shown in Figure 7 in line 676-681, the expression levels of all tested genes in qRT-PCR were consistent with FPKM from RNA-seq.

Question 9: Line 122: it should be "based on"

Answer: We are sorry for the error. The sentence was corrected in the revised manuscript and see line 122.

Question 10: Line 156: were the differences significant?

Answer: Thank you for your question. The median of gene expression levels was significant difference between the samples treated with different light quality ($p < 0.05$) and see line 159-161.

Question 11: Line 159: what does mean "for simples"?

Answer: We are very sorry for spell mistake. It should be "for samples" and it was modified in line 163.

Question 12: Line 234: Delete the word "in"

Answer: It was deleted the word "in" in the revised manuscript.

Question 13: Line 250: In which tissue?

Answer: Thank you for your comments. RL promoted lycopene and β -carotene accumulation in grapefruit pulp and we have added relevant information in the manuscript in line 259.

Question 14: Line 320: should be: "positively correlated"

Answer: Thank you for point this out. We revised it in the manuscript in line 329.

Question 15: Line 332: should be: "responds"

Answer: It was corrected, see line 343.

Question 16: Figure 3: colors of the Venn diagrams are slightly misleading. Try to use stick to the colors of your light treatments and use red, blue, black and grey (for white), or - do not use colors at all.

Answer: We are sorry for our problem. In figure 3B, red circle and blue circle, respectively, represent 184 and 220 days after blossom and we have supplement information in figure 3 legend in line 654-655. Therefore, it is difficult to distinguish developmental stages for grapefruit if we use the color of light treatments, namely red, blue, black and grey (for white), or do not use color at all.

Question 17: Figure 8. Explain all abbreviations used in the figure description.

Answer: We accepted this suggestion. We added full name of all abbreviations in the figure 8 legend in line 690-697.

Question 18: The Table S7 is not cited in the main text.

Answer: We are sorry for it. The Table S9 (namely Table S7 in previous manuscript) was cited in the materials and methods in the revised manuscript, see line 448.

Question 19: Provide the reference number of the grapefruit reference genome which was used for the gene annotation here.

Answer: We accepted this suggestion. Up to now, because of no grapefruit genome available, we choose genome of sweet orange (*Citrus sinensis* L.), the congeneric fruit for grapefruit, as reference genome. Sweet orange genome can be found under the accession AJPS000000000 (version: GCF_000317415.1_Csi_valencia_1.0) in NCBI database and seen line 417-418.

Xu, Q. et al. The draft genome of sweet orange (*Citrus sinensis*). *Nat. Genet.* **45**, 59–66 (2013).

REVIEWERS' COMMENTS:

Reviewer #1 (Remarks to the Author):

My comments were properly responded.

but there were some commmets:

Line 253 `ζ-carotene lycopene ` should be `ζ-carotene, lycopene `.

Line 271 'These results showed that red light play important regulatory role in carotenoid metabolism of grapefruit.' These were the results of red light-transmittance bagging, not the results of red light.

Line 356 'a preliminary regulatory model of red light-induced carotenoid metabolism' This was the result of red light-transmittance bagging, not the result of red light.

Reviewer #3 (Remarks to the Author):

Dear Authors,

Compared to its previous version, I find the revised manuscript largely improved. You have professionally addressed all minor and major issues. Especially, I would acknowledge complementing the current manuscript with the data from the natural light experiment as well as with the real-time qPCR analysis.

Additionally, I have found very few minor issues, which should be quickly corrected before publication. While reading the revised manuscript, I have spotted several typing mistakes (f.e.: correct into "Relative" in Table 1C or "verification" in line 346), therefore I would encourage the Authors to double-check the manuscript text throughout or use professional proofreading service. The manuscript would certainly benefit from adding the information about the method used for calculation of relative expression levels (in Materials and methods, Real-time quantitative PCR section). Was the Livak method (Livak and Schmittgen, 2001) or the Pfaffl method (Pfaffl, M.W., 2001. A new mathematical model for relative quantification in real-time RTPCR. Nucleic Acids Res. 29, 16–21.) used? Also, as the real-time PCR reactions were performed with the SYBR Green dye, it would be beneficial to add the information about the specificity of real-time PCR reactions (namely: was the melting curve/ dissociation curve analysis done for each amplification product?). Including all the above-mentioned information into your manuscript would certainly further improve it, however, to my opinion the manuscript is ready for publication in its current form.

Reviewer #1 (Remarks to the Author):

My comments were properly responded.

but there were some commmets:

Question 1: Line 253 ‘ ζ -carotene lycopene ‘ should be ‘ ζ -carotene, lycopene ‘.

Answer: We are sorry for the error. The sentence was corrected in the revised manuscript and see line 250.

Question 2: Line 271 ‘These results showed that red light play important regulatory role in carotenoid metabolism of grapefruit.’ These were the results of red light-transmittance bagging, not the results of red light.

Answer: Thank you for point this out. It was revised in line 267 of the manuscript.

Question 3: Line 356 ‘a preliminary regulatory model of red light-induced carotenoid metabolism’ This was the result of red light-transmittance bagging, not the result of red light.

Answer: Thank you very much, it was revised, see line 347-349.

Reviewer #3 (Remarks to the Author):

Dear Authors,

Compared to its previous version, I find the revised manuscript largely improved. You have professionally addressed all minor and major issues. Especially, I would acknowledge complementing the current manuscript with the data from the natural light experiment as well as with the real-time qPCR analysis.

Question 1: Additionally, I have found very few minor issues, which should be quickly corrected before publication. While reading the revised manuscript, I have spotted several typing mistakes (f.e.: correct into “Relative” in Table 1C or “verification” in line 346), therefore I would encourage the Authors to double-check the manuscript text throughout or use professional proofreading service.

Answer: Thank you for point this out. We revised it in the manuscript in line 346 and supplementary information. In addition, we double-checked the manuscript text throughout.

Question 2: The manuscript would certainly benefit from adding the information about the method used for calculation of relative expression levels (in Materials and methods, Real-time quantitative PCR section). Was the Livak method (Livak and Schmittgen, 2001) or the Pfaffl method (Pfaffl, M.W., 2001. A new mathematical model for relative quantification in real-time RT-PCR. *Nucleic Acids Res.* 29, 16–21.) used? Also, as the real-time PCR reactions were performed with the SYBR Green dye, it would be beneficial to add the information about the specificity of real-time PCR reactions (namely: was the melting curve/ dissociation curve analysis done for each amplification product?).

Answer: Thank you for your comments. The relative expression of the genes was calculated according to the $2^{-\Delta\Delta CT}$ method (Livak and Schmittgen, 2001) and seen in line 463-464. Melting curve analysis was done to ensure the specificity for each amplification product see line 462-463.

Livak, K. J. & Schmittgen, T. D. Analysis of relative gene expression data using real-time quantitative PCR and the $2^{-\Delta\Delta CT}$ method. *Methods* **25**, 402–408 (2001).